# Corrosion Behaviors of Fe-22Cr-16Mn-0.55N High-Nitrogen Austenitic Stainless Steel in 3.5% NaCl Solution

**Song Xu** [1], **Fengyin Gao** [2], **Jianyang Han** [2], **Shangfeng Xiong** [1], **Xinyu Duan** [1], **Fanglin Zha** [1], **Bing Yu** [1], **Lanlan Yang** [2], **Yanxin Qiao** [2,*], **Zhibin Zheng** [3,*] **and Jian Chen** [4]

1   Electric Power Research Institute of State Grid Hunan Electric Power Co., Ltd., Changsha 410007, China
2   School of Materials Science and Engineering, Jiangsu University of Science and Technology, Zhenjiang 212003, China
3   Institute of New Materials, Guangdong Academy of Sciences, National Engineering Research Center of Powder Metallurgy of Titanium & Rare Metals, Guangdong Provincial Key Laboratory of Metal Toughening Technology and Application, Guangzhou 510650, China
4   Department of Chemistry, Western University, London, ON N6A 5B7, Canada
*   Correspondence: yxqiao@just.edu.cn (Y.Q.); zbzheng712003@163.com (Z.Z.)

**Abstract:** In this study, the corrosion behavior of the high-nitrogen austenitic stainless steel (HNS) Fe-22Cr-16Mn-0.55N before and after solution treatment (ST) in 3.5% NaCl solution has been investigated. The effect of a solution temperature of 1100 °C and heat preservation for 30 min on the corrosion resistance and passive film protection of HNS steel was studied. Open-circuit potential, potentiodynamic polarization and electrochemical impedance tests were used to assess the corrosion resistance of treated and untreated steels. In addition, potentiostatic polarization and XPS techniques together with Mott–Schottky curves were applied to determine the composition and properties of the passive films. The results showed that after solution treatment, the grain size of HNS decreased and the grain became more uniform. Although corrosion occurred on HNSs both before and after treatment, solution treatment resulted in greater compactness in passive films, leading to lower carrier density, lower corrosion current density and better corrosion resistance.

**Keywords:** high nitrogen austenitic stainless steel; solution treatment; corrosion; passive film

## 1. Introduction

Austenitic stainless steels (ASSs) are widely used in many environments, such as pipeline transportation, construction, chemical industry, etc., due to their excellent all-round properties [1–5]. However, ASSs are vulnerable to corrosion damage, especially pitting, in corrosive environments [6–8]. Pitting holes are generally small in diameter, but they can also be deep with unpredictable formation patterns. Pitting corrosion can destroy passive films with serious consequences [9,10]. For this reason, many methods are used to improve the corrosion resistance of ASSs [11,12].

Solution treatment is a relatively mature technology which can reduce second-phase precipitation and improve the mechanical properties and corrosion resistance of materials. Zhou et al. [13] found that HNS in a mixed-acid solution of 0.8 mol/L $H_2SO_4$ and 0.8 mol/L HCl exhibited enhanced corrosion resistance after appropriate heat treatment. Second-phase stainless steel contains $M_{23}C_6$ carbides (where M represents Fe, Cr or Mo), as well as intermetallic compounds [14,15] which enable the structural components of stainless steel to become uniformly distributed, so that any harmful phase can be fully dissolved into the solid solution and re-precipitation is inhibited [16]. Hao et al. [17] reported that the banded microstructure formed in austenitic stainless steel after cold and hot rolling was eliminated by solution treatment at 1100 °C for 30 min. In solution-treated stainless steel, carbides and inclusions entirely dissolve in the austenite matrix, and rapid cooling is used to obtain a single austenite structure at room temperature which is less susceptible to intergranular

corrosion [18]. The occurrence of intergranular corrosion is usually due to unevenness inside the grains [19]. The actual mechanism of intergranular corrosion has been well explained by the theory of poor chromium, which states that rapid cooling of stainless steel after high-temperature heating causes supersaturated C to form $Cr_{23}C_6$ and precipitate from the grain boundary. The precipitation of $Cr_{23}C_6$ and the slow diffusion of Cr in the austenite crystal mean that Cr is not replenished sufficiently quickly, resulting in Cr depletion at the grain boundary. Intergranular corrosion then occurs due to the formation of corrosion cells between grains in Cr-depleted regions that are prone to corrosion [20].

The precipitation of $Cr_{23}C_6$ greatly reduces the content of chromium at the grain boundary and causes the depletion of Cr at the grain boundaries [21]. During solution treatment, the required temperature for nitride precipitation is not attained because of rapid cooling. This means there is no nitride precipitation phase after solution treatment, resulting in greatly improved corrosion resistance. The solution treatment counters the influence of the precipitation phase in the high-nitrogen steel and results in an even distribution of the nitrogen element.

In this study, we investigated the corrosion properties of HNS before and after solution treatment in 3.5% NaCl solution. We analyzed open-circuit potential, potentiodynamic polarization and electrochemical impedance to assess the corrosion resistance of treated and untreated steels. We also used potentiostatic polarization and XPS techniques and carried out analyses of Mott–Schottky curves to determine the composition and properties of the passive films.

## 2. Experimental Details

### 2.1. Materials

The experimental material used in this work is a high-nitrogen austenitic stainless steel (HNS) (Fe-22Cr-16Mn-0.55N mass fraction, %) with content including 0.044 C, 0.24 Si, 16.84 Mn, 0.017 P, 0.005 S, 22.40 Cr, 0.55 N, 1.50 Ni, and Fe. It was provided by Northeast University (Shenyang, China). The CALPHAD approach was employed to calculate the fractions of equilibrium phases in the HNS by means of Pandat TM software (demo 2017) [22]. The simulated equilibrium diagram of the HNS is shown in Figure 1. We subjected the steel to solution treatment at 1100 °C for 30 min and then cooled the material to room temperature using water. We refer to the original high nitrogen austenitic stainless steel as HNS. For HNS subjected to solution treatment, we use the term ST.

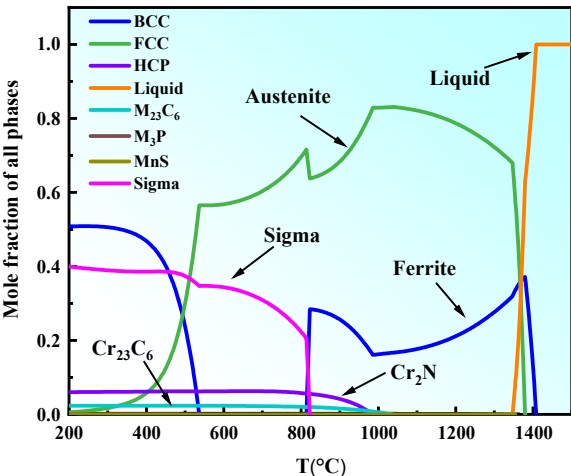

**Figure 1.** Equilibrium fraction plot for each phase of HNS versus temperature.

### 2.2. Sample Preparation

Prior to the experiment, HNS and ST were into specimens with dimensions of 10 mm × 5 mm × 5 mm. We mechanically ground samples with 400#, 1000# and 2000# sandpaper and polished them roughly and finely. The surfaces were cleaned with ultrasonic



waves and alcohol, then air-dried before use. The samples were electro-etched in $FeCl_3$ solution (10 g $FeCl_3$, 30 mL HCl, and 120 mL $H_2O$) for 10 s. Electron backscatter diffraction (EBSD) (Oxford Instruments Technology Corporation, Oxford, UK) was used to observe the microstructure of the test steel, and XL30-FEG scanning electron microscopy (SEM, FEI, Hillsboro, OR, USA) was used to observe the corrosion morphology of the test steel.

### 2.3. Experimental Method

We performed electrochemical tests using a Reference 600+ electrochemical workstation (Gamry Instruments, Inc., Philadelphia, PA, USA). We used a traditional three-electrode system with the test sample as the working electrode, a platinum plate as the auxiliary electrode and a saturated calomel electrode (SCE) as the reference electrode. The tested solution is 3.5% NaCl, which was prepared by using distilled water and NaCl in a completely degassed state.

Prior to the potentiodynamic polarization curve test, we carried out open-circuit potential (OCP) measurements of samples for 3600 s to ensure that the system reached a steady state. We then performed open-circuit potential electrochemical impedance spectroscopy (EIS) with a test frequency range of $10^4$–$10^{-2}$ Hz and an amplitude of 10 mV. We then measured the potentiodynamic polarization curves. Potential sweep ranges for HNS and ST were −0.5 V below OCP and termed when the current density attains 10 mA/cm$^2$ with the sweep rate was 0.333 mV/s. The potential of the potentiostatic polarization test was −0.14 V and the duration time was 1 h. After potentiostatic polarization, the samples were subjected to Mott–Schottky testing immediately. The test potential range was −0.6~1.2 V, the frequency was $10^3$ Hz, the amplitude was 10 mV and continuous step size was 30 mV. We carried out all experiments at room temperature (25 ± 1) °C at least three times.

The sample size is 10 mm × 10 mm × 2 mm. First, we conducted mechanical polishing to 2000#, then mechanical polishing to remove scratches, and finally stress relief polishing with 0.5-particle-size $SiO_2$ abrasive solution. The microstructure and Electron Backscattered Diffraction (EBSD) was conducted using scanning electron microscopy (GeminiSEM, Zeiss, Oberkochen, Germeny). The EBSD test was conducted using an Oxford Instruments Nordlys Nano detector (Oxford Instruments Technology Corporation, Oxford, UK) with step size of 0.5 µm and the obtained EBSD data were analyzed using the software HKL Channel 5. X-ray photoelectron spectroscopy analysis (XPS, ESCALAB 250Xi T, ThermoFisher Scientific, Waltham, MA, USA) was conducted to characterize the composition of the passive film formed at 0 V vs. SCE. This was performed using an Al $K_\alpha$ (1486.6 eV) radiation source. The obtained XPS data were analyzed using CasaXPS peak (version 2.3.15) fitting software.

## 3. Results and Discussion

### 3.1. EBSD Analysis

Figure 2a,b shows the grain orientation diagrams for HNS and ST. The normal directions of the samples are shown in colors which correspond to the statistics of grain size. Before solution treatment, the grains are larger and mostly irregular in shape, with larger grain boundaries. After solution treatment, a recrystallization process occurs, leading to the regular and uniform equiaxed grains [23]. Figure 2c,d show calculated grain sizes of HNS and ST. The average HNS grain size was about 50 µm. The average ST grain size was about 18 µm. The authors of [24] found that the smaller the grain size, the more uniform the particles, and the better the corrosion resistance. Figure 2e,f show the KAM maps of HNS and ST. Figure 2e shows that HNS exhibits a higher stress state, which can be attributed to the high dislocation density of the material. These help to block the connectivity of high-angle grain boundaries and prevent the expansion of corrosion cracks associated with such boundaries. To a certain extent, this reduces the corrosion tendency of HNS and results in improved corrosion resistance [25].

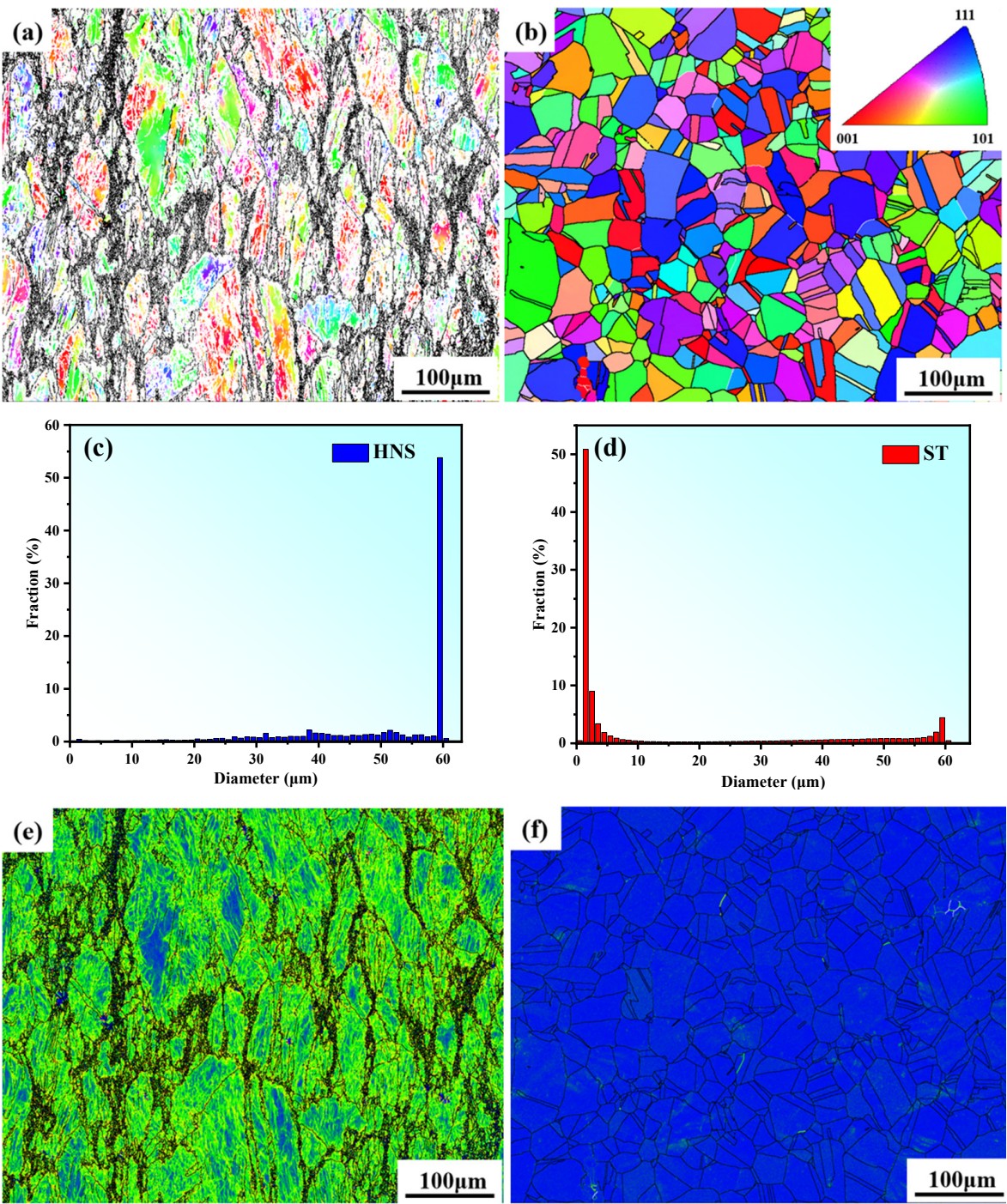

**Figure 2.** Inverse pole figures of (**a**) HNS, (**b**) ST; grain size distribution of (**c**) HNS, (**d**) ST; KAM diagrams of (**e**) HNS, (**f**) ST.

*3.2. Electrochemical Tests*

Figure 3 shows the OCP responses of HNS and ST in 3.5% NaCl solution. Both steels exhibit a positive shift trend, but the HNS trend does not rise over time, while the ST trend rises sharply at first, and then continues to rise steadily to levels above those of HNS, and finally reaches a stable level. These results show that HNS struggles to form a relatively dense passive film, while such film readily forms on the surface of ST, which is in a spontaneous passive state [26,27]. The OCP of ST is higher than that of HNS, while the corrosion tendency of ST is slightly lower [28,29].

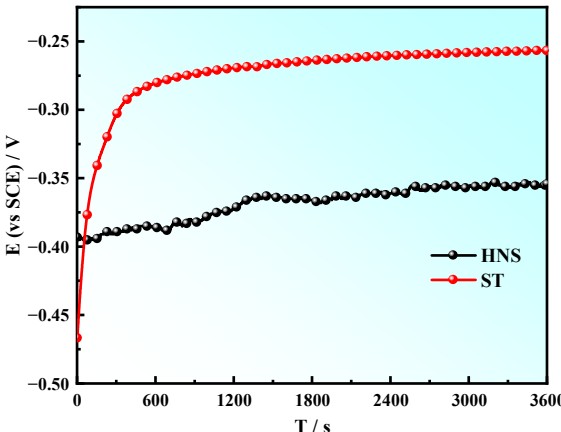

**Figure 3.** OCP responses of HNS and ST in 3.5% NaCl solution.

Figure 4 shows the potentiodynamic polarization curves of HNS and ST steels. In both cases, the passive region is small, but the ST region is the larger of the two, indicating that the passive film stability of ST is higher than that of HNS, a finding which is consistent with the open-circuit unit result. Table 1 lists the corrosion potential ($E_{corr}$), corrosion current density ($i_{corr}$) and pitting potential ($E_P$) values obtained from Figure 4. The corrosion potential of ST is slightly higher than that of HNS. After solution treatment, the corrosion current density of high-nitrogen austenitic stainless steel falls from $1.12 \times 10^{-7}$ A·cm$^{-2}$ to $0.25 \times 10^{-7}$ A·cm$^{-2}$, while pitting potential rises from $-0.12$ V$_{SCE}$ to $-0.03$ V$_{SCE}$. These results show that solution treatment can improve the corrosion resistance of high-nitrogen austenitic stainless steel. Chen et al. [30] observed that when the solution temperature increased from 800 °C to 1100 °C, Cr$_2$N was gradually eliminated, and corrosion resistance was greatly improved.

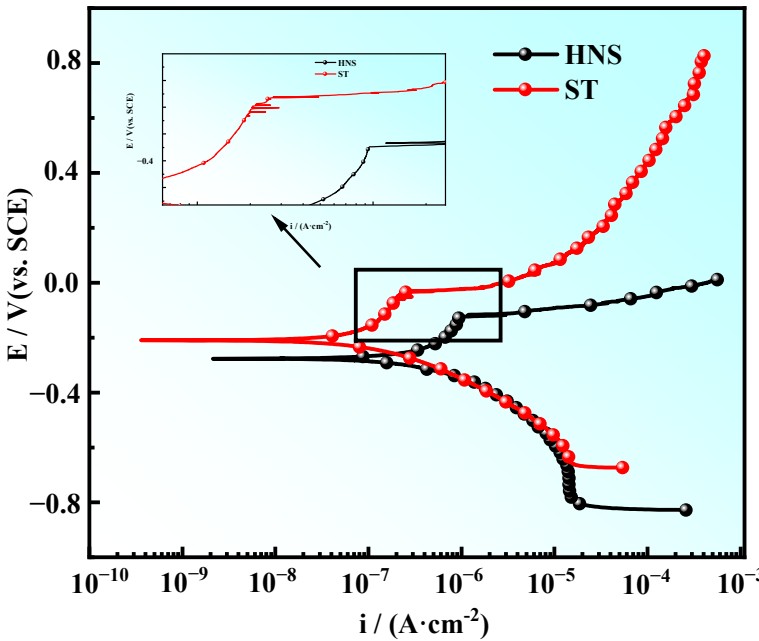

**Figure 4.** Potentiodynamic polarization curves for HNS and ST in 3.5% NaCl solution.

**Table 1.** Fitting results of the polarization curves for HNS and ST.

| Samples | $E_{corr}$/V$_{SCE}$ | $i_{corr}$/A·cm$^{-2}$ | $E_p$/V$_{SCE}$ |
|---------|---------|---------|---------|
| HNS | $-0.27 \pm 0.02$ | $(1.12 \pm 0.10) \times 10^{-7}$ | $-0.12 \pm 0.02$ |
| ST | $-0.21 \pm 0.02$ | $(0.25 \pm 0.12) \times 10^{-7}$ | $-0.03 \pm 0.01$ |

Figure 5 shows the surface morphologies of HNS and ST after the potentiodynamic polarization test. The corrosion degree of the two test steels is uniformly set at 10 A cm$^{-2}$ current density. We found corrosion in both materials in the form of corrosion pits which formed during the corrosion process of the second-phase Cr$_2$N. The Figure 5b,d show that the corrosion pits on HNS are larger, deeper, and more extensive than those on ST. This shows that corrosion resistance of the tested steel is significantly improved by solution treatment.

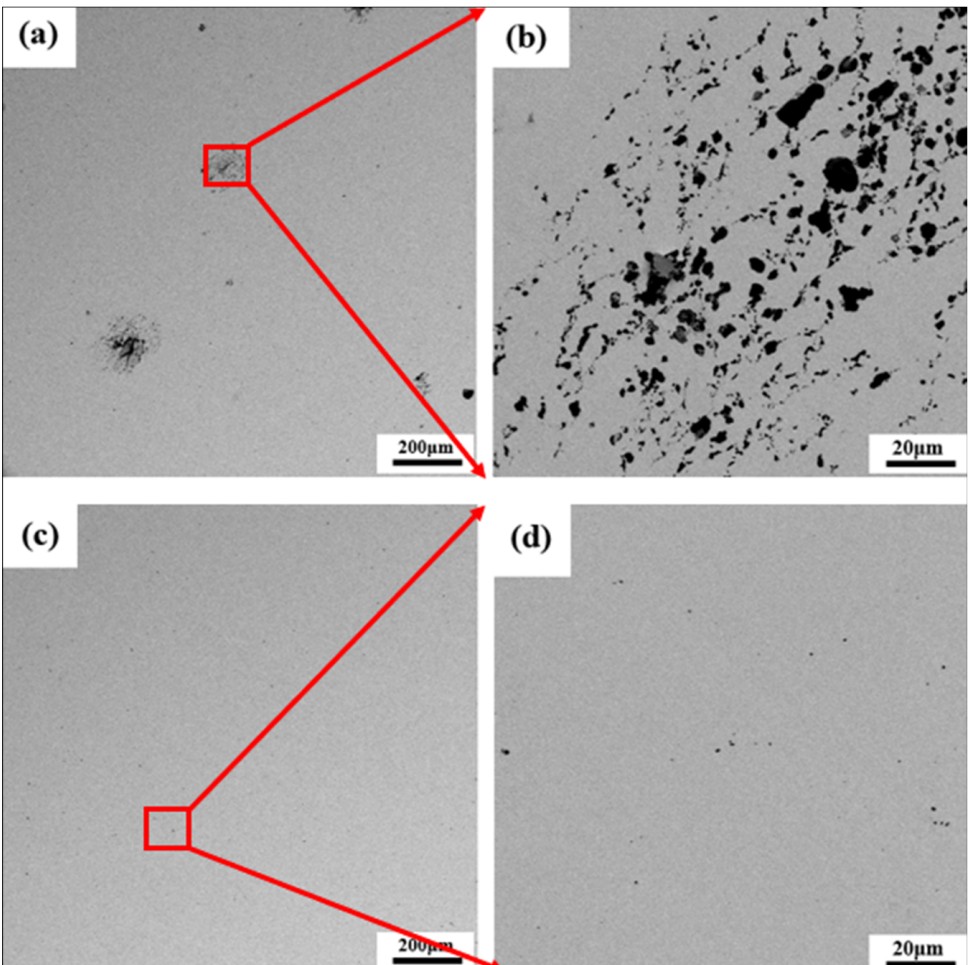

**Figure 5.** Surface morphologies of corroded HNS (**a**) corrosion morphology, (**b**) high magnification of the red square in Figure 5a, and ST (**c**) corrosion morphology, (**d**) magnification of the red square in Figure 5c.

Figure 6 shows EIS plots for HNS and ST. The Nyquist plot reveals incomplete semi-circular arcs, indicating capacitance behavior [31,32]. Our results show that the corrosion response of Fe-22Cr-16Mn-0.55N steel treated in 3.5% NaCl solution is similar to that previously found for Fe-18Cr-15Mn-0.66N [33] and for Fe-18Cr-15Mn-2Mo-0.66N [34]. In Figure 6a, the arc curvature radius of HNS is smaller than that of ST, indicating that ST has superior corrosion resistance [35]. The Bode plot shows that the ST phase angle tends to 80° in the frequency range of approximately 10$^{-1}$ to 10$^3$ Hz, suggesting that the passive film has some protective qualities [36]. The phase angle of HNS is lower, indicating poorer protective performance of the passive film [37].

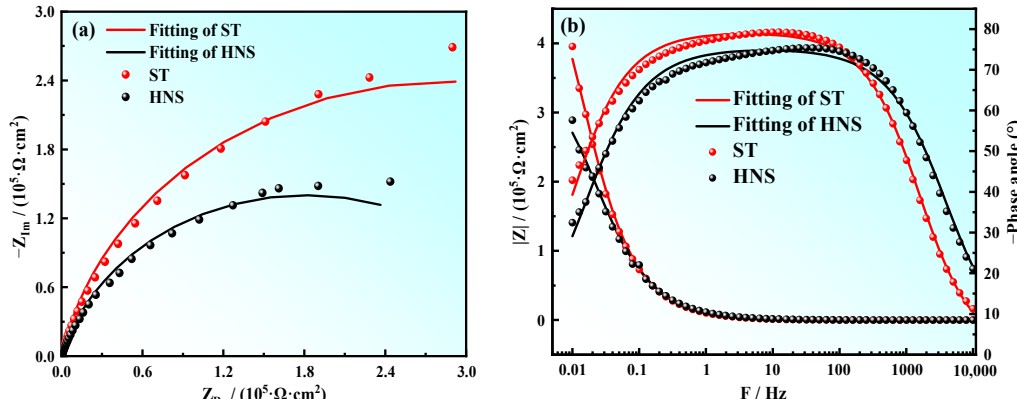

**Figure 6.** (**a**) Nyquist plot; and (**b**) Bode plot of electrochemical impedance spectroscopic data of HNS and ST.

Figure 7 shows the equivalent circuit diagram used to fit the AC impedance spectrum. In this figure, $R_s$ is the solution resistance, $R_p$ is the resistance of the passive film and CPE is the constant phase element. We calculated impedance as follows:

$$Z_{CPE} = \frac{1}{Y_0}(j\omega)^{-n} \tag{1}$$

where $Y_0$ is a proportionality constant, $j_2 = -1$ and $n$ is an empirical exponent between 0 and 1. When $n = 1$, CPE represents a pure capacitive element; when $n = 0$, CPE represents a resistor; and when $n = 0.5$, CPE represents Warburg impedance [38].

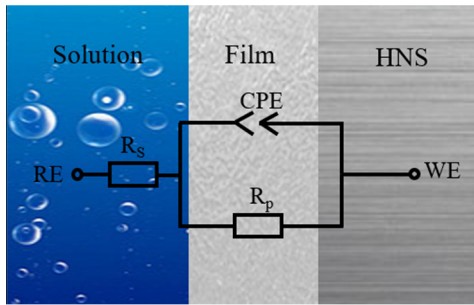

**Figure 7.** The equivalent circuit used for quantitative evaluation of the EIS. (RE is reference electrode, $R_s$ is the solution resistance, $R_p$ is the resistance of the passive film, CPE is the constant phase element and WE is working electrode).

Table 2 lists $R_{ct}$ values for the two materials. $R_s$ and $R_p$ values increase after solution treatment. This indicates that solution treatment results in the formation of a more stable and protective passive film with better corrosion resistance.

**Table 2.** EIS fitted results for HNS and ST.

| Samples | $R_s/\Omega \cdot cm^2$ | $Q/\Omega^{-1} s^n cm^{-2}$ | $n$ | $R_p/\Omega \cdot cm^2$ |
|---------|-------------------------|------------------------------|-----|--------------------------|
| HNS | 11.24 | $1.88 \times 10^{-5}$ | 0.84 | $3.63 \times 10^5$ |
| ST | 15.53 | $1.96 \times 10^{-5}$ | 0.88 | $5.76 \times 10^5$ |

### 3.3. Passive Film Performance

3.3.1. Potentiostatic Polarization

Figure 8 shows the current–time transients of HNS and ST obtained at $-0.14$ V potential. Figure 8a shows that the steady-state current density of HNS is high, while the steady-state current density of ST is low and relatively stable at $5 \times 10^{-8}$ (A·cm$^{-2}$). These

findings show that solution treatment enhances the protection offered by the steel passive film to the substrate [39].

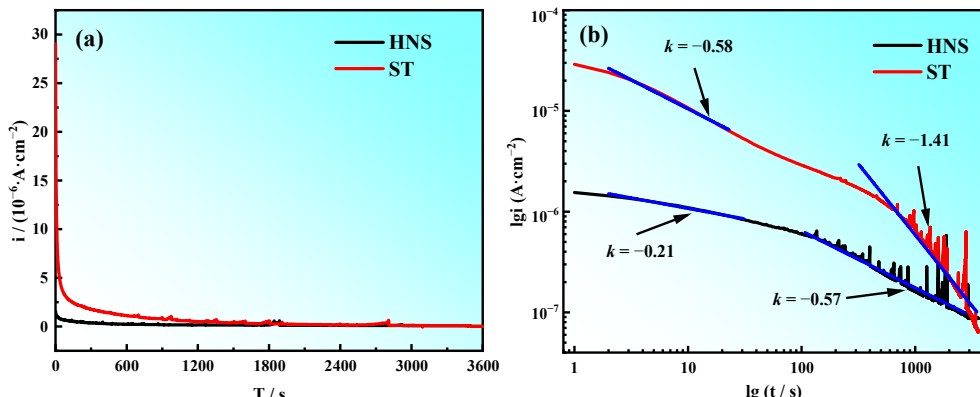

**Figure 8.** (**a**) Current–time curves; and (**b**) double-log plots of current–time for HNS and ST.

We express time dependency of the current response as follows:

$$i = 10^{(-(A+k\log t))} \tag{2}$$

where $i$ is current density, $t$ is time and $A$ is a constant. We obtain slope $k$ from the double log plot in Figure 8b. The slopes of the double logarithmic current density–time transient plot $k$ represent the growth rate of the passive films [32]. When $k = -1$ indicates that the growth process is controlled by the electric field, a denser and protective passive film is formed; $k = -0.5$ indicates that a porous film is formed due to the dissolution and precipitation process. Both curves are composed of two linear regions. The first segment ($t < 100$ s) represents the passivation film formed in air, while the second segment ($t > 100$ s) represents the passivation film formed in 3.5% NaCl solution. The $k$ value of HNS decreased from $-0.21$ to $-0.57$; the $k$ value of ST decreased from $-0.58$ to $-1.41$. It shows that the two test steels are porous at the beginning, and the passive film of ST is denser and more protective at the later stage. It shows that solution treatment with a solution temperature of 1100 °C and a heat preserve of 30 min increases the density of the HNS passive film and increases corrosion resistance.

### 3.3.2. Mott–Schottky Analysis

Figure 9 shows Mott–Schottky curves for HNS and ST under $-0.1$ $V_{SCE}$. The Mott–Schottky plots exhibit two obvious linear regions (I and II). In region I, the negative slope of the straight line suggests p-type semiconducting behavior in the passive film. The positive slope of the straight line in region II suggests n-type semiconducting behavior [40]. This is due to the presence of $Fe_2O_3$, $Fe_3O_4$ and FeOOH in the passive film.

We determine the donor density ($N_D$) of passive films by means of linear fitting using the following equation (for n-type semiconductors):

$$\frac{1}{C^2} = \frac{2}{\varepsilon \varepsilon_0 e N_D}\left(E - E_{FB} - \frac{kT}{e}\right) \tag{3}$$

where $\varepsilon$ is the dielectric constant of the film unaffected by hydrogen (15.6 F/cm), $\varepsilon_0$ is the dielectric constant of the oxide film ($8.854 \times 10^{-14}$ F/cm) and $e$ is the electron charge ($1.602 \times 10^{-19}$ C). $E$ is the applied potential, $E_{FB}$ is flat-band potential, $k$ is Boltzmann constant ($1.38 \times 10^{-23}$ J/K) and $T$ is the temperature [41].

Table 3 shows values of donor density ($N_D$) obtained from Figure 9. ST has a lower $N_D$ value than HNS. This finding shows that high-nitrogen austenitic stainless steel forms a more protective passive film after solution treatment in 3.5% NaCl solution, thereby improving the corrosion resistance of the material [42]. Therefore, the heat treatment

process with a solution temperature of 1100 °C and a heat preserve of 30 min can improve HNS with poor corrosion resistance. This result is also consistent with the impedance fitting results.

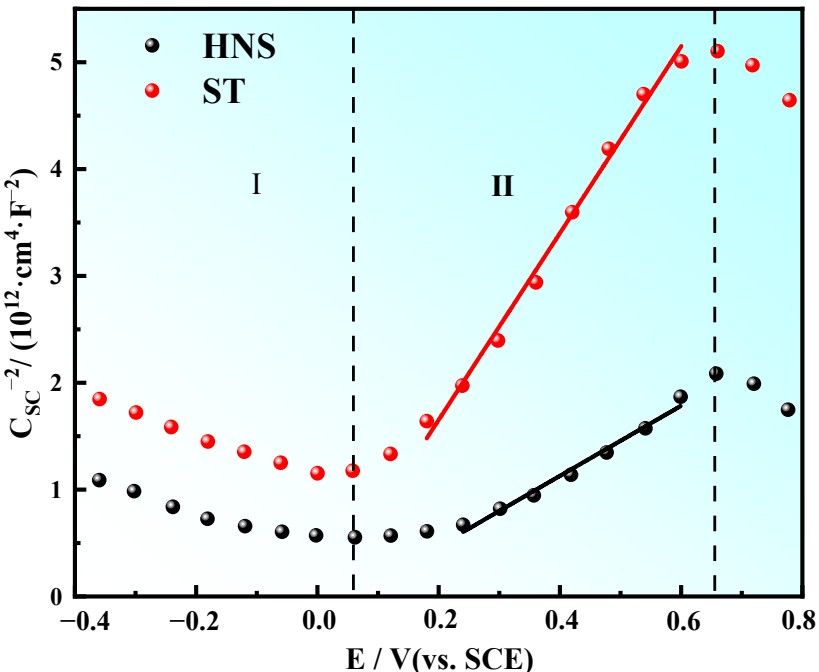

**Figure 9.** Mott-Schottky curves for HNS and ST.

**Table 3.** Donor densities of passive films formed on HNS and ST.

| Samples | HNS | ST |
|---|---|---|
| $N_D$ (cm$^{-3}$) | $2.76 \times 10^{18}$ | $1.03 \times 10^{18}$ |

### 3.4. Passive Film Anylysis

The stability and protection performance of passive film is most affected by its composition. Figure 10 shows the XPS spectra of O1s, Cr2p, Fe2p and N1s in the passive films of the two materials. Figure 10a,b shows fitting results of the O1s spectra in the passive films of HNS and ST. For both states of steel, the O element exhibits three peaks, as follows: $(529.9 \pm 0.1)$ eV for $O^{2-}$, $(531.4 \pm 0.1)$ eV for $OH^-$ and $(532.6 \pm 0.1)$ eV for $H_2O$. For HNS, the $O^{2-}/OH^-$ ratio is 0.62. For ST, the corresponding ratio is 0.67. An increased $O^{2-}/OH^-$ ratio results in improved protective performance of the passive film and reduced pitting corrosion [43].

Figure 10c,d show the XPS spectra of Cr 2p3/2 in the passive films of HNS and ST. For both states of steel, there are three peaks of the Cr element, as follows: $(574.6 \pm 0.3)$ eV for Cr, $(576.6 \pm 0.2)$ eV for $Cr_2O_3$ and $(578.1 \pm 0.3)$ eV for $Cr(OH)_3$. Compared with HNS, ST has a higher content of Cr and $Cr_2O_3$, indicating that solution treatment promotes the enrichment of Cr in the passive film of high-nitrogen steel, and the enrichment of Cr is of great importance for the formation of oxides.

Figure 10e,f shows the XPS spectra of Fe 2p3/2 in the passive films of HNS and ST. Before and after solution treatment, the Fe element exhibits four peaks, as follows: $(707.9 \pm 0.3)$ eV for metallic Fe, $(709.3 \pm 0.3)$ eV for $Fe_3O_4$, $(710.7 \pm 0.1)$ eV for $Fe_2O_3$ and $(712.7 \pm 0.4)$ eV FeOOH. The metallic Fe content is lower in HNS while the oxides $Fe_3O_4$, $Fe_2O_3$ and FeOOH that make up the passive film of Fe are higher in ST, and this increase in Fe oxide and FeOOH content makes the passive film more protective.

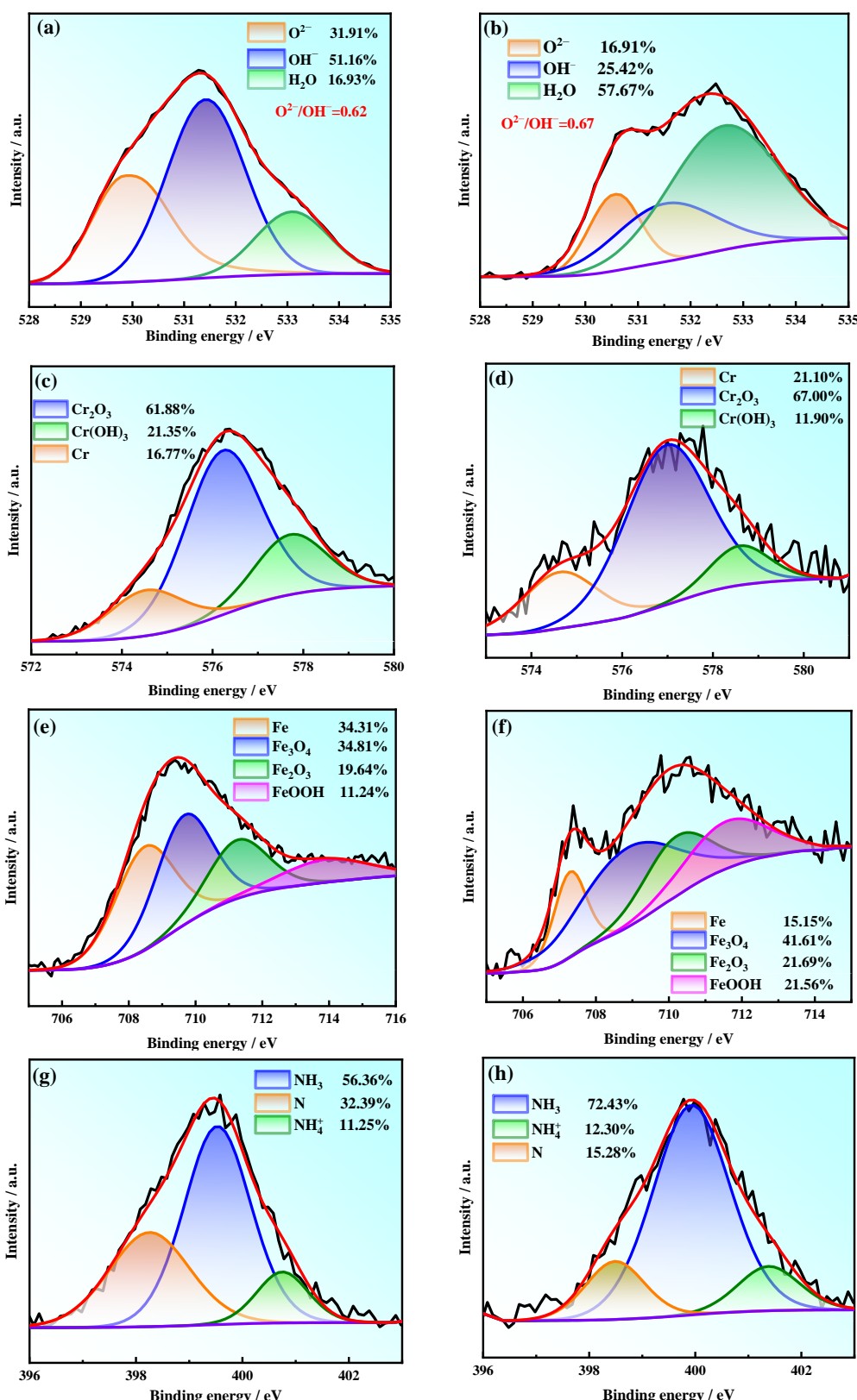

**Figure 10.** XPS spectra of O1s, Cr2p, Fe2p and N1s in HNS (**a,c,e,g**) and ST (**b,d,f,h**) passive films. Black line is the tested dada and the red line is the dada obtain form CasaXPS peak analysis.

Figure 10g,h show the XPS spectra of N1s in the passive films of HNS and ST. Before and after solution treatment, the N element exhibits three peaks for free N, $NH_3$, and $NH_4^+$, respectively. A comparison of the two figures shows that the free N in the passive

film decreases after solution treatment, while the content of $NH_4^+$ increases. $NH_4^+$ inhibits rupturing of the passive film and reduces the likelihood of pitting corrosion [44].

This helps to explain why the passive film of solid-solution high-nitrogen steel remains stable during potentiostatic polarization, while the passive film of forged high-nitrogen steel becomes highly unstable. We can state, therefore, that solution treatment with a solution temperature of 1100 °C and a heat preserve of 30 min improves the protective ability of the passive film on the HNS surface to the substrate, thereby improving the corrosion resistance of the material.

### 4. Conclusions

(1) Compared with HNS, the test steel at solution temperature shows higher open circuit potential, lower activity, lower corrosion current density and superior corrosion resistance.

(2) The steady-state current density of HNS was very high, and its passive film was extremely unstable. After solid solution, the current density was low, and the passive film was relatively stable and dense. The passive films of both materials exhibited n-p structures. However, the carrier density of ST passive film was lower than that of HNS, resulting in superior protection.

(3) The concentration of Cr in the ST passive film was relatively large, resulting in improved protective ability. The content of Fe metal declined in ST, while the content of the oxides $Fe_2O_3$ and FeOOH increased, and this also improved the protective ability of the passive film.

(4) The solution treatment process (solution temperature of 1100 °C, solution time of 30 min) set this time has greatly improved the corrosion resistance of HNS and the passive film protection of HNS.

**Author Contributions:** Data curation, S.X. (Song Xu), F.G. and J.H.; Writing—original draft, S.X. (Song Xu), F.G., J.H. and Z.Z.; Writing—review & editing, S.X. (Shangfeng Xiong), X.D., F.Z., B.Y., L.Y., Y.Q. and J.C. All authors have read and agreed to the published version of the manuscript.

**Funding:** This research was funded by the Science and Technology Project of the State Grid Hunan Electric Power Company (No. 5216A520002T), the National Key Research and Development Program of China (No. 2021YFB3701704), GDAS' Project of Science and Technology Development (No. 2022GDASZH-2022010103).

**Institutional Review Board Statement:** Not applicable.

**Informed Consent Statement:** Not applicable.

**Data Availability Statement:** Not applicable.

**Conflicts of Interest:** The authors declare no conflict of interest.

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
