# Peer review of "Corrosion Behaviors of Fe-22Cr-16Mn-0.55N High-Nitrogen Austenitic Stainless Steel in 3.5% NaCl Solution"

_coatings, doi:10.3390/coatings12111769_

Round 1
Reviewer 1 Report
All the acronyms should be defined before their first appearance in text
Some English formulation are quite poor for example sentence from line 18 is not clear at all. Please revise the entire manuscript with an appropriate English native or professional
I expected to see some results in brief in abstract
“demanding environments” is a large industrial field – please be specific
The introduction require a better structure and more critical review as now the state of art were very briefly presented
The scientific knowledge presented in this work are quite limited- therefore not clear which can be authors contribution to improve the state of art
The chemical analysis were performed by authors or these data were taken from supplier or literature data ?
The quality of Figures is very poor especially for Figure 1, 2 5 as the resolution is low and there are missing essential detail as scale bar and the figures itself do not allows a proper interpretation of data indicated
Details of sample preparation for ebesd and measurement details should be presented in details
Table 1 presents its chemical composition are not representative as low and por quality/resolution
The statistical meaning of EBSD results are questionable as not large area were investigated
“We found corrosion in both materials” – where are some trace of corrosion in your Figure 5 ? for me it is not clear !
Some quantitative details are required in conclusion – now they are very qualitative and not have much impact
Author Response
Dear Reviewer:
This is my review feedback.
1、The application site of "demanding environments" has been specified.
2、Experimental materials and chemical analysis provided by Northeast University (Shenyang, Liaoning).
3、Figures 1, 2 and 5 have been modified.
4、Details of sample preparation for ebesd and measurement details has been presented in details.
5、Table 1 Chemical composition has been deleted. Only the contents of several main elements are introduced.
6、Figure 5 Corrosion diagram has been modified in detail.
7、This paper focuses on the comparison of HNS corrosion resistance improvement with solution temperature of 1100 ℃ and solution time of 30min.
Thank you!

Reviewer 2 Report
This is a peer review of Corrosion Behaviours of Fe-22Cr-16Mn-0.55N High Nitrogen Austenitic Stainless Steel in 3.5 % NaCl Solution.
Basically the authors followed a standard suite of tests but never explained what do the results mean? Do they recommend this steel? In which conditions? For what applications? Why? Tests were conducted in one electrolyte at one temperature only. The obtained data is not compared to anything in the literature.
First of all it is difficult to assess the purpose of this research. The authors studied only one steel under two thermal treatment conditions in a salt solution but never compared their data to the most common austenitic steels such as type 304SS. What was the aim of this research? Is this Mn containing steel supposed to be a replacement for 304SS? if yes, you need to compare both alloys.
The authors never mention is the 3.5% NaCl solution was deaerated or not. This will give totally different corrosion potentials. Are the plots in Figures 3,4&6 representative values? How did you know which one of the three repetitions to publish?
Figures 9&10 for example show data. But the authors never explain how this data is useful in interpreting the experimental data obtained? Does this data help in recommending the Mn containing steel?
The figure captions need to be more complete. For example Figure 5 shows images of "corroded" steels. Corroded how? Just plain immersion tests? for how long? in which electrolyte? etc.
Figure 7 is very nice. Thank you!
Authors use correctly the word specimen only once. But they use the word sample several times. These are not synonym words. Only specimens can be tested. A specimen is prepared from a representative sample of material.
The authors use words like "deep" "very high" "extremely" etc. that are unquantifiable. Better give numbers or correlations.
Author Response
Deer Reviewer:
The following is the feedback of the revised paper.
1、The purpose of this study is to design a solution treatment process (solution temperature 1100 ℃, solution time 30 min) to improve the corrosion resistance of HNS and the protection of passive film, and compare with the original sample before solution treatment.
2、3.5% NaCl solution has been degassed.
3、Figures 3, 4 and 6 have been repeated for many times, and the repeatability is very high, so a representative group is selected.
4、Figure 9 studies the performance of the passive film. The carrier density is obtained through calculation. Figure 10 obtains the content of each substance in the passive film through XPS. By comparing the two test steels, it is found that the solution treatment process we designed (solution temperature 1100 ℃, solution time 30 min) has significantly improved the protection of the passive film.
5、The corrosion morphology scanned in Fig. 5 has been corrected. And the experiment is completed under 10mA · cm-2 through polarization curve in 3.5% NaCl to ensure that the corrosion resistance of the two materials is observed under the same corrosion condition.
6、The word "sample" has been modified.

Reviewer 3 Report
Manuscript is devoted to the comprehensive study corrosion properties of high nitrogen austenitic stainless with and without solution treatment steel in 3.5 % NaCl solution. It contains results obtained by many methods, including Open-circuit potential, potentiodynamic polarization and electrochemical impedance tests, potentiostatic polarization, XPS and Mott–Schottky curves.
Line 94. It is stated that “Prior to the potentiodynamic polarization curve test, we carried out open-circuit potential (OCP) measurements of samples for 3600 s to ensure that the system reached a steady state”. Though Fig. 3 does not show that potential of ST sample is stable after 3600 s.
Potentiostatic studies were not carried out correctly, therefore the description of Fig. 8 and conclusion (3) are also not correct. The samples were kept at a constant potential of –0.05 V, which is lower than the pitting potential of –0.03 V for ST steel, but higher than the pitting potential of –0.12 V for HNS steel (Table 2). I.e., the samples were in incomparable states: a passive film was formed on the ST steel, and the HNS steel was already dissolving in the pores, so the HNS steel was in a known more corrosive state. Therefore, authors can either (a) remove this piece of research from the manuscript, or (b) change the description and conclusions, or (c) perform potentiostatic exposure at the values of the potentials of the passivation region for both samples.
Manuscript can be published after major revision.
Author Response
Deer Reviewer:
Here is my paper feedback.
1、The OCP curve has been modified.
2、The potential selection of constant potential is modified and the curve is corrected.
Thank you!

Reviewer 4 Report
I want to note that the most successful part of the article is the determination of the electrochemical properties of the surface of the steel under study. However, the authors do not quite correctly capture the mechanism of corrosion failure. Everything revolves around the well-known hypothesis about the occurrence of intergranular cracks due to the depletion of grain boundaries in chromium and precipitates of chromium carbide of the type Cr23C6.
Based on chemical analysis, we see that nitrogen in the hoist (Table 1) is an order of magnitude greater than carbon. In addition, nitrogen has a greater affinity for iron and chromium than carbon. It follows from this that iron and chromium nitrides must first form, then carbonitrides of these elements. And the formation of carbide Cr23C6. may not be enough carbon. Поэтому
1. It is necessary to carry out a detailed phase analysis of the investigated steel.
2. The photos shown in Fig. 5 are not very informative, the captions to them are also not visible. Their interpretation is weak, so you need to either redo it or completely remove it.
3. Conclusions 2 and 4 require clarification and argumentation. They should not be written in the same words as in the text of the article.
4. Austenitic steels are usually poisoned with "aqua regia", the authors use an etchant for steels of the ferritic or ferrite-martensitic class. Maybe that's why the quality of the above photos.
Author Response
Deer Reviewer:
This is my paper revision feedback.
1、Design of solution treatment process determined by thermal equilibrium phase diagram.
2、Figure 5 has been modified to correct the corrosion morphology.
3、The conclusion has also been revised to determine the center of the paper.
4、To modify the etchant, FeCl3 solution (10g FeCl3, 30ml HCl, and 120ml H2O) is selected.

Round 2
Reviewer 1 Report
.
Author Response
Thank your for your kind helo.
Reviewer 2 Report
Thank you for your answers. I wish you would copy (this is not too difficult to do) the comments and answer below. Seeing the answers without the actual question or comments is like hearing only one side of a phone conversation.
I was also not able to see how was the response to other reviewers. But this maybe a problem of the journal, not of the authors.
Author Response
Thank your for your kind helo.
Reviewer 3 Report
A big work has been done with this manuscript. Photos of surface morphology (Fig. 5) are significantly improved. Though some changes in the manuscript raised new questions.
Significant difference between OCP responses of ST in 3.5% NaCl solution in old and in new versions of Figs. 3 (after 1 hour of exposure the potential difference is almost 100 mV!) indicate the irreproducibility of either the result of the solution treatment steel, or the preparation of the steel surface before electrochemical studies. This, in turn, casts doubt on the results of all subsequent electrochemical studies.
In addition to this:
The new version of Fig. 8 shows the current–time transients of HNS and ST obtained at -0.14 V potential, but it shows the same the current–time transient of HNS that was presented in the old version and it was obtained at –0.05 V potential. Moreover, the segments of the same dependence have somewhat modified slopes in the new edition. This further strengthens the distrust of the results obtained by the authors.
Therefore, I cannot recommend this manuscript for publication.
Author Response
Significant difference between OCP responses of ST in 3.5% NaCl solution in old and in new versions of Figs. 3 (after 1 hour of exposure the potential difference is almost 100 mV!) indicate the irreproducibility of either the result of the solution treatment steel, or the preparation of the steel surface before electrochemical studies. This, in turn, casts doubt on the results of all subsequent electrochemical studies.
Response:
Thank you for your valuable comments. The passive/oxide film formed on the surface of the sample in open air has a significant influence on the open corrosion potential (OCP) response of stainless steel in OCP test. In this work, the difference of the previous test may due to the original state to the composition of the passive/oxide film. The detailed description of the properties of the passive/oxide film on the corrosion properties of metallic materials can be found in the published work [1].
[1] Z.R. Ye, Z.C. Qiu, Z.B. Wang, Y.G. Zheng, R. Yi, X. Zhou, Can the Prior Cathodic Polarisation Treatment Remove the Air-Formed Surface Film and Is It Necessary for the Potentiodynamic Polarisation Test?, Acta Metallurgica Sinica (English Letters) 33(6) (2020) 839-845.
In addition to this:
The new version of Fig. 8 shows the current–time transients of HNS and ST obtained at -0.14 V potential, but it shows the same the current–time transient of HNS that was presented in the old version and it was obtained at –0.05 V potential. Moreover, the segments of the same dependence have somewhat modified slopes in the new edition. This further strengthens the distrust of the results obtained by the authors.
Response:
Thank you for your valuable comments. The results of this work showed that both the as received HNS and ST HNS has inferior pitting resistance especially as received HNS. The microstructure especially inclusions, such as Cr2N has deleterious effect on localized corrosion of HNS [1]. As seen in Fig. 5 (a) and (b), there exitance of plenty of tiny corrosion pits, this may due to the Cr2N and/or non-metallic inclusions such as (Mn, Cr)-oxide. During the potentiostatic polarization test, the I-t response is closely related to the properties of the passive film on the sample surface especially at the sites of the properties of Cr2N and/or (Mn, Cr)-oxide. In the revised work, the I-t response in Fig. 8 (a) showed the passive behavior of HNS, the this may due to the microstructure.
The current–time transients in Fig. 8 indicting the occurrence of the pitting corrosion and unstable of the passive film. This is frequently observed in potentiostatic polarization test for stainless steel in Cl- contained solution.
As to the modified slopes in the new edition, this is a common knowledge. In the revised version, the potentiostatic polarization test conducted in -0.14V, in the previous test, the potentiostatic polarization test conduced in -0.05V. So the formation dynamic of the passive is different to each other, if the slope is the same as that of the previous work, I this this may be impossible.
In fact, The reproducibility of electrochemical experiments is not really very dood. At last, I insist the results provided is true and reliable.
[1] H.Y. Ha, H.S. Kwon, Effects of Cr2N on the pitting corrosion of high nitrogen stainless steels, Electrochimica Acta 52(5) (2007) 2175-2180.
Reviewer 4 Report
After corrections and additions, the article can be published.
Author Response
Thank your for your kind help.
Round 3
Reviewer 3 Report
Please find comments attached.

Author Response
Dear reviewers:
- In version v1 of the manuscript (Fig. 3), the stationary potential of ST steel after 3600 s of exposure was about -0.16 V, and in versions v2 and v3 it was about -0.26 V for the same ST steel. That is, a consequence of the irreproducibility of the result for treated steel ST potential difference is about 100 mV.
The non-reproducibility can be caused both by the non-reproducibility of the results of the treatment of the HNS steel itself to obtain ST steel, and the non-reproducibility of the surface samples preparation before measurements. In this case, the difference in corrosion potentials when comparing the corrosion resistance of HNS and ST steels is about 60 mV (Table 1 and Fig. 4), i.e., it is significantly less than 100 mV.
Since the stationary potential largely determines the properties of the surface, with such a low reproducibility it is not correct to conclude that SUCH steel treatment is advantageous, but only about the advantage of a specific SINGLE sample (it has no scientific significance and practical meaning).
Therefore, before carrying out electrochemical studies, it is necessary to achieve reproducibility of the stationary potential of the treated steel.
Response:
Thank you for your valuable comments. According to the electrochemical response especially the potentiodynamic polarization curves presented in Fig. 4, the HNS used in this work may be a completely failure. But a detailed analysis of the HNS can help us to derive the reasons for failure. It may provide some insight to other scholars to avoid making the same mistake.
- The red curves (ST steel) are exactly the same in all three versions, including all current surges. At the same time:
1) The signed slopes in version 1 are DIFFERENT from version 2 and 3;
2) It is stated that the potential at which the dependence in version 1 was obtained is (–0.05 V); in version 2 it is (–0.14 V), in version 3 it is again (–0.05 V).
The black curve (HNS steel) is exactly the same in version 2 and 3, but in version 2 it is indicated that it was obtained at –0.14V, and in version 3 it was obtained at –0.05V.
Response:
Thank you for your valuable comments. We sincerely apologize to you, that we made a very big mistake in the revision 2. The I-t response of the Solution Treated HNS we used the wrong data. In revision 2, the raw data of Solution Treated HNS is the same as the first version. Based on your comments raised, we remeasurements the electrochemical test many times, but Unfortunately we use the wrong data. This has caused your misunderstanding. The I-t curves obtained in the revision test was conducted in -0.14V. The revision data were listed below.
Once again, we sincerely apologize for the mistakes for the inconvenience caused by our manuscript.
Yanxin Qiao
on behalf of all the authors.